# Unveiling the Metabolic Trajectory of Pig Feces Across Different Ages and Senescence

**DOI:** 10.3390/metabo14100558

**Published:** 2024-10-17

**Authors:** Chuanmin Qiao, Chengzhong Liu, Ruipei Ding, Shumei Wang, Maozhang He

**Affiliations:** 1Hainan Provincial Key Laboratory of Tropical Animal Breeding and Disease Research, Institute of Animal Science and Veterinary Medicine, Hainan Academy of Agricultural Sciences, Haikou 571199, China; 2School of Basic Medical Sciences, Anhui Medical University, Hefei 230032, China; 2013270014@stu.ahmu.edu.cn (C.L.); 2113100010@stu.ahmu.edu.cn (R.D.); 2018530001@ahmu.edu.cn (S.W.)

**Keywords:** swine, aging, feces, metabolomics

## Abstract

Porcine models are increasingly recognized for their similarities to humans and have been utilized in disease modeling and organ grafting research. While extensive metabolomics studies have been conducted in swine, primarily focusing on conventional cohorts or specific animal models, the composition and functions of fecal metabolites in pigs across different age groups—particularly in the elderly—remain inadequately understood. In this study, an untargeted metabolomics approach was employed to analyze the fecal metabolomes of pigs at three distinct age stages: young (one year), middle-aged (four years), and elderly (eight years). The objective was to elucidate age-associated changes in metabolite composition and functionality under standardized rearing conditions. The untargeted metabolomic analysis revealed a diverse array of age-related metabolites. Notably, L-methionine sulfoxide levels were found to increase with age, whereas cytidine-5-monophosphate levels exhibited a gradual decline throughout the aging process. These metabolites demonstrated alterations across various biological pathways, including energy metabolism, pyrimidine metabolism, lipid metabolism, and amino acid metabolism. Collectively, the identified key metabolites, such as L-methionine sulfoxide and Cholecalciferol, may serve as potential biomarkers of senescence, providing valuable insights into the mechanistic understanding of aging in pigs.

## 1. Introduction

The influence of aging on porcine production and health is a critical area of study that not only enhances livestock management but also contributes to our understanding of human aging and disease [1,2]. As pigs age, they experience physiological changes that can affect growth, reproduction, and immune function, leading to implications for meat quality and overall health. By studying these age-related transformations in pigs, researchers can gain valuable insights into the biological processes of aging that are also relevant to humans. This comparative approach allows for a better understanding of age-related diseases, such as metabolic disorders and immunosenescence in both species. Consequently, findings from porcine aging studies can inform strategies for improving health outcomes in humans, highlighting the interconnectedness of animal and human health in a One Health framework. Aging is a complex biological process involving the interaction of various molecular and cellular mechanisms, such as theories of programmed longevity, free radical immunity, wear and tear, cross-linking, endocrine, and basal metabolism to describe the aging process [3,4]. Despite significant advances in aging research over the past decades, a comprehensive understanding of metabolic changes during the aging process remains limited. In recent years, an increasing number of researchers have begun to utilize omics technologies to study normal aging due to their high-throughput characteristics in proteomics, genomics, transcriptomics, and metabolomics [5,6,7,8]. Metabolomics, as a branch of systems biology, offers a powerful tool for the study of aging by quantifying all metabolites within a biological system [9].

In the field of agricultural science, understanding the aging process in animal models such as pigs is of significant importance for comprehending aging and age-related diseases, as well as optimizing breeding management strategies to enhance production efficiency and meat quality [10]. Pigs, as an important agricultural animal model with physiological and anatomical characteristics similar to humans, serve as an ideal subject for studying aging-related metabolic changes [11]. With the rapid development of metabolomics technologies, especially the application of high-resolution mass spectrometry and nuclear magnetic resonance techniques, studying the metabolic changes in pigs at the molecular level has become feasible [12,13]. The aim of this study is to utilize metabolomics methods to analyze the metabolite profiles of pigs of different age groups to identify metabolic changes associated with aging. By conducting non-targeted metabolomics analysis on blood and tissue samples from young, middle-aged, and elderly pigs, we aim to reveal age-related metabolites and their changes in biological pathways. This information will not only aid in understanding the aging mechanisms in pigs but also provide a reference for aging research in other species, including humans.

In this study, we initially collected feces samples from pigs of three different age groups: (1) one year old, (2) four years old, and (3) eight years old [14]. Subsequently, non-targeted metabolomics analysis was performed using liquid chromatography-mass spectrometry (LC-MS) techniques. Through data analysis, we identified a series of age-related metabolites and observed significant metabolic changes in multiple biological pathways, including pyrimidine metabolism, vitamin metabolism, and amino acid metabolism, which play key roles in the aging process. Furthermore, we found that these metabolic changes are associated with known aging markers such as oxidative stress, inflammation, and cellular damage. These findings suggest that the changes in fecal metabolites can serve as a powerful tool to uncover molecular changes during the aging process in pigs, leading to the development of new biomarkers and therapeutic targets for age-related diseases. It may also provide new targets for the development of strategies to delay aging or improve animal welfare and human health.

## 2. Materials and Methods

### 2.1. Study Design and Sample Collection

A cohort of 45 pigs (sex: female; breed: Tunchang pig, Hainan province, China; Body weight: approximately 92~108 kg) was enrolled in this study on 21–22 August 2022, with fresh fecal samples collected from each individual. The pigs were categorized into three age groups, comprising 15 pigs each: one year old, approximately four years old, and approximately eight years old. This study design facilitated the collection of comprehensive and representative biological samples across various age points, which is essential for investigating the complexities of the aging process.

### 2.2. Chemicals and Regents

Sigma Aldrich (Darmstadt, Germany) provided ammonium acetate (NH_4_AC) and HPLC-grade formic acid and HPLC-grade ammonium formate. Merck provided acetonitrile (Darmstadt, Germany). Thermo Fisher (Waltham, MA, USA) provided ammonium hydroxide (NH_4_OH), methanol, MS-grade methanol, MS-grade acetonitrile, and HPLC-grade 2-propanol. All chemicals used in this study were of high purity and quality, ensuring the reliability and reproducibility of the results.

### 2.3. Metabolomics Study of Feces Samples

The feces samples were placed in the EP tubes and resuspended with prechilled 80% methanol by well vortex. Then, the samples were melted on ice and whirled for 30 s. After the sonification for 6 min, they were centrifuged at 5000 rpm, 4 °C for 1 min. The supernatant was freeze-dried and dissolved with 10% methanol. Finally, the solution was injected into the LC-MS/MS system analysis. UHPLC (ultra-high-performance liquid chromatography)-MS/MS analyses were performed using a Vanquish UHPLC system (Thermo Fisher, Waltham, MA, USA) coupled with an Orbitrap Q Exactive^TM^ HF mass spectrometer (Thermo Fisher, Waltham, MA, USA) in Novogene Co., Ltd. (Beijing, China). Samples were injected onto a Hypesil Gold column (100 mm × 2.1 mm, 1.9 μm, Thermo Fisher Scientific, Waltham, MA, USA) using a 12 min linear gradient at a flow rate of 0.2 mL/min. The eluents for the positive polarity mode were eluent A (0.1% FA in Water) and eluent B (Methanol). The eluents for the negative polarity mode were eluent A (5 mM ammonium acetate, pH 9.0) and eluent B (Methanol). The solvent gradient was set as follows: 2% B, 1.5 min; 2–85% B, 3 min; 85–100% B, 10 min; 100–2% B, 10.1 min; 2% B, 12 min. Q ExactiveTM HF mass spectrometer (Orbitrap MS, Thermo Fisher Scientific, Waltham, MA, USA) was operated in positive/negative polarity mode with a spray voltage of 3.5 kV, capillary temperature of 320 °C, sheath gas flow rate of 35 psi and aux gas flow rate of 10 L/min, S-lens RF level of 60, and aux gas heater temperature of 350 °C. Compound Discoverer 3.1 (CD3.1, Thermo Fisher, Waltham, MA, USA) was used to perform peak alignment, peak picking, and metabolite quantification after the raw data files were generated by UPLC-MS/MS. The primary parameters used in this analysis included a retention time tolerance of 0.2 min, an actual mass tolerance of 5 ppm, a signal intensity tolerance of 30%, a signal-to-noise ratio of 3, and a minimum intensity threshold of 100,000. Peak intensities were normalized to the total spectral intensity and utilized to predict molecular formulas based on additive ions, molecular ion peaks, and fragment ions. These peaks were then matched with the mzCloud (https://www.mzcloud.org/, accessed on 7 April 2020) and HMDB (https://hmdb.ca/, accessed on 7 January 2022), LipidMaps (http://www.lipidmaps.org/, accessed on 10 November 2023) databases for annotation. Consequently, accurate qualitative and relative quantitative results were obtained. Partial least squares discriminant analysis (PLS-DA) was employed to elucidate the metabolic changes among the groups using the R package ropls. Metabolites demonstrating variable importance in projection (VIP) greater than 1 and a corrected *p*-value (Wilcoxon test) less than 0.05 were identified as differential metabolites and then subjected to enrichment analysis. The enrichment pathways of the differential plasma metabolite profiles between any two groups of pigs were analyzed using MetaboAnalyst 5.0 (http://www.metaboanalyst.ca, accessed on 15 January 2021); when the *p*-value of metabolic pathway < 0.05, the metabolic pathway was considered statistically significant enrichment.

### 2.4. Statistical Analysis

All statistical analyses were conducted using the R platform (version 4.0). For comparisons among multiple groups, a one-way ANOVA was employed to evaluate differences among the three groups. Only indices that demonstrated significant differences were further assessed using the Mann–Whitney U test with Bonferroni correction as a post hoc analysis between each pair of groups. Adjusted *p*-values of less than 0.05 were considered statistically significant. Error bars represent the mean ± standard error (SE). Principal component analysis (PCA) was performed to investigate the difference in the metabolome among different age groups via the vegan package of R software (Version 4.2.2, Auckland). Following PCA analyses, The PerMANOVA (permutational multivariate analysis of variance) analysis was conducted to evaluate the difference among different age groups using the Adonis function in the vegan package of R software, and *p*-values less than 0.05 were regarded as statistically significant.

## 3. Results

### 3.1. The Overall Characteristics of Feces Metabolomics

In metabolomics studies utilizing mass spectrometry, quality control (QC) samples are routinely employed to ensure the reliability and high quality of the acquired metabolomics data. Although QC samples are theoretically identical, systematic errors are often introduced during the processes of sample extraction, detection, and analysis. These errors can lead to variations among QC samples; the smaller the variations, the higher the method stability and data quality.

In the present study, the primary aim is to investigate the trajectory of changes in fecal metabolic profiles across various age stages in pigs, specifically at one year of age (Ad), four years of age (Ma), and eight years of age (Oa) (Figure 1A). Principal component analysis (PCA) revealed a significant separation among the three age groups using PERMANOVA (*p* = 0.002, and 0.014), indicating distinct metabolic profiles associated with aging. Additionally, the analysis highlighted a tight clustering of the QC samples, which suggests a high degree of stability and repeatability in the detection process. This dense distribution of QC samples reinforces the reliability of the obtained data (Figure 1B,C).

### 3.2. The Trajectory of Dynamic Changes of Fecal Metabolites Associated with Aging

In the differential metabolomics analyses, a substantial number of metabolites were identified as significantly distinct among the swine in the middle-aged (Ma, four years old) group compared to the one-year-old (Ad) group, the elderly (Oa, eight years old) group compared to the Ad group, and the Oa group compared to the Ma group. Volcano plots indicated that a total of 383, 268, and 225 differentially expressed metabolites were detected in these comparisons, respectively (Appendix A). Among these metabolites, the upregulated counts were 333, 225, and 60, while the downregulated counts were 50, 43, and 192, respectively (Figure 2A–F).

In the positive ion mode, pairwise comparisons between the three groups revealed 53, 42, and 71 group-specific differential metabolites for the Ma vs. Ad, Oa vs. Ad, and Oa vs. Ma comparisons, respectively. Additionally, eight common differential metabolites were identified across these comparisons (Figure 2G). In contrast, a total of 38, 21, and 34 group-specific differential metabolites were identified in the negative ion mode; however, no overlapping differential metabolites were found in this mode (Figure 2H). The eight common differential metabolites identified include 4-(acetylamino) phenyl 3-chlorobenzoate, amoxicillin, 1-(4-nitrophenyl) piperidine, cyanidin, 4-guanidinobutanoic acid, ethyl 5-methoxy-2-methyl-1-phenyl-1H-indole-3-carboxylate, 5-(tert-butyl)-2-methyl-N-(4-nitrophenyl)-3-furamide, and valine.

### 3.3. Screening of Fecal Metabolites Intimately Linked to Aging

Differential metabolites were identified based on three key parameters: variable importance in projection (VIP), fold change (FC), and *p*-value. To qualify as differential metabolites, a VIP value greater than 1 and a *p*-value less than 0.05 were required. As a result, metabolites were selected as differentially expressed according to the criteria of a *p*-value below 0.05 and a VIP value exceeding 1, with FC values either greater than 1 or less than 1. In pairwise comparisons among the one-year-old (Ad), four-year-old (Ma), and eight-year-old (Oa) groups, the top 20 upregulated and downregulated metabolites exhibiting the highest fold changes were selected for analysis. In the comparison between the Ma and Ad groups, 12,13-EODE demonstrated the highest level of upregulation, while N-(5-acetamidopentyl) acetamide showed the most significant downregulation (Figure 3A). In the comparison between the Oa and Ad groups, methyl 2,5-dimethyl-AH-pyrrole-3-carboxylate exhibited the greatest upregulation, whereas QQH was the most pronounced downregulated metabolite (Figure 3B). Finally, in the comparison between the Oa and Ma groups, 4-hydroxyindole was identified as the most upregulated metabolite, while 3-(G-nitropyridin-2-yl)oxy-1H-indazole exhibited the highest level of downregulation (Figure 3C). Collectively, these findings highlight distinct metabolic alterations associated with aging and underscore the potential relevance of these metabolites in understanding the biochemical changes that occur across different life stages in pigs.

As the age of the pig population increases, 2-thio-acetyl MAGE, cholecalciferol, and L-methionine sulfoxide have been observed to exhibit an increasing trend (Figure 4A–C), while pyridoxamine, 2-deoxyuridine, and cytidine-5-monophosphate have shown a decreasing trend (Figure 4D–F). The identification of these down-regulated metabolites may provide insights into metabolic pathways that become less active over time. Together, these visual representations facilitate a clearer understanding of the dynamic metabolic shifts that accompany the aging process in pigs, underscoring the importance of both up-regulated and down-regulated metabolites in age-related research.

### 3.4. KEGG Enrichment Analysis Based on Differential Metabolites

In the comparative analysis between the Ma (four years old) and Ad (one year old) groups, a total of 15 metabolic pathways were identified. These pathways predominantly included those related to vitamin digestion and absorption, taurine and hypotaurine metabolism, prolactin signaling, antifolate resistance, sphingolipid signaling, lysine degradation, linoleic acid metabolism, galactose metabolism, D-arginine and D-ornithine metabolism, the cGMP-PKG signaling pathway, and adrenergic signaling in cardiomyocytes.

When comparing the Oa (eight years old) group with the Ad (one year old) group, 12 metabolic pathways were identified, including metabolic pathways, cysteine and methionine metabolism, renin secretion, regulation of lipolysis in adipocytes, prolactin signaling pathway, cAMP signaling pathway, antifolate resistance, adrenergic signaling in cardiomyocytes, vitamin digestion and absorption, pyrimidine metabolism, vitamin B6 metabolism, and steroid biosynthesis.

In the comparison between the Oa (eight years old) and Ma (four years old) groups, three metabolic pathways were identified, including Pyrimidine metabolism, ABC transporters, and beta-Alanine metabolism (Figure 5A,B).

Furthermore, the analysis revealed that pyrimidine metabolism and metabolic pathways are significantly differential profiles common to all three groups (Appendix A). Pyrimidine metabolism plays a crucial role in cell division and DNA synthesis and is closely associated with the aging process. As age progresses, the rate of cellular division slows, and the capacity for DNA repair declines, which can affect the normal progression of pyrimidine metabolism. Abnormalities in pyrimidine metabolism may lead to the accumulation of abnormal metabolites within cells, potentially impacting cellular function and even leading to cell death.

## 4. Discussion

This study leverages metabolomics to uncover and identify differences in metabolite composition among pigs aged 1 (Ad), 4 (Ma), and 8 years old (Oa). Results indicate that metabolites such as cholecalciferol (vitamin D), L-methionine sulfoxide, pyridoxamine, and cytidine-5-monophosphate exhibit monotonic changes. For instance, L-methionine sulfoxide and cholecalciferol gradually increase with the age of the pigs, whereas the levels of cytidine-5-monophosphate decrease progressively as the pigs age. The KEGG enrichment analysis uncovered several critical metabolic pathways that were significantly differentially regulated during the aging process in pigs. Notably, these pathways included vitamin digestion and absorption, cysteine and methionine metabolism, vitamin B6 metabolism, and pyrimidine metabolism. The prominence of these pathways suggests that they play vital roles in the biochemical shifts associated with aging, potentially influencing overall metabolic health and physiological function in aging pigs [15].

Cholecalciferol, also known as vitamin D, plays a multitude of crucial roles in the human body, including the regulation of calcium and phosphorus metabolism and bone health [16]. Adequate levels of vitamin D are essential for maintaining bone strength and preventing osteoporosis. As age advances, the body’s requirement for vitamin D increases in order to maintain skeletal health, and deficiency is associated with an increased risk of osteoporosis and fractures in the elderly [17]. Previous studies have suggested that vitamin D may be related to various physiological processes, including immune function, cell growth and differentiation, and endocrine regulation, all of which are closely related to the aging process [18]. Therefore, maintaining appropriate vitamin D levels may help delay certain aspects of aging, such as the decline in muscle strength and cognitive function [19]. Aging is a natural and inevitable process of molecular and cellular damage accumulation, leading to functional deficits in cells, tissues, and entire organs, thereby weakening the entire body [20]. With the decline in overall immune capacity during aging, the relative number of senescent immune cells decreases [21], reducing immune surveillance, such as the detection and destruction of tumor cells by cytotoxic T cells [22]. Individuals with low immune function have a significantly higher risk of cancer than those with high immune function [23]. The cancer-protective effect of adequate Vitamin D status may be primarily related to the ability to maintain a high level of immune capacity [24]. Additionally, Souraya et al. reported that vitamin D can enhance immunity during the aging process by partially blocking p38 MAPK signaling, thereby inhibiting the production of SASP (senescence-associated secretory phenotype) in senescent cells [25]. Our study found that cholecalciferol showed an increasing trend with age, contrasting with the decrease typically seen in human aging, suggesting that its sufficiency is an essential component of healthy aging in swine, not only helping to maintain the good condition of bones and skeletal muscles but also contributing to the homeostasis of the immune system of elderly pigs, consistent with the views of Fantini et al. [26]. However, this difference between pigs and humans highlights the need for species-specific research in aging. Understanding these variations can provide insights into how vitamin D metabolism impacts health in different species.

Next, L-methionine sulfoxide (MetO) is a form of protein that has undergone a change in function and structure due to oxidation by reactive oxygen species (ROS) and hydrogen peroxide when exposed to methionine and methionyl residues under physiological or pathological conditions. In mammals, oxidative stress is usually accompanied by an inflammatory response, generating ROS and other molecules that cause tissue damage. Reports indicate that MetO can promote the initiation of inflammatory responses [27]. Lee et al. proposed that the reduction of L-methionine sulfoxide promotes mitophagy, which is involved in the process of mitophagy [28]. Excessive ROS can damage mitochondria, and mitophagy can remove damaged mitochondria, protecting cells from apoptosis. Furthermore, the role of L-methionine sulfoxide in health and disease has been studied in various organisms from bacteria to mammals, including humans. For example, in mammals, elevated levels of MetO are involved in the expression of markers related to neurodegenerative diseases [29], mental health disorders [30], hearing loss [31], cystic fibrosis lung disease [32], macular degeneration [33], cardiovascular diseases [34], liver and kidney toxicity [35], and cancer [36]. Our study shows that MetO also exhibits an increasing trend with age. Catanesi et al. found that a diet rich in MetO may demonstrate therapeutic potential for human aging and age-related diseases, protecting neurons from oxidative imbalance and maintaining mitochondrial function [37]. In the present study, it was demonstrated that levels of L-methionine sulfoxide (MetO) gradually increase with porcine aging. This accumulation of MetO may elevate the risk of neurodegenerative diseases in swine, such as Alzheimer’s disease (AD) [38]. The increased levels of MetO could adversely affect critical cellular processes, including mitophagy, which is essential for maintaining mitochondrial health and function. Impaired mitophagy is closely associated with cellular aging and apoptosis, both of which significantly contribute to the pathogenesis of age-related disorders. Utilizing pigs as a model allows for further investigation into how MetO accumulation influences neurodegenerative processes, exploration of the involved molecular pathways, and examination of potential interventions. The findings from this study provide valuable insights into the effects of MetO on porcine aging, underscoring the relevance of pig models in advancing research on aging and related diseases.

Aging and age-related diseases, such as diabetes, atherosclerosis, and neurodegenerative diseases, are characterized by an increase in oxidative chemical modification of tissue proteins [39]. Glyoxal or advanced glycation end-products (AGEs) are produced by secondary modification of proteins by carbohydrate oxidation products and are associated with the severity of diabetic complications in the kidneys, retina, and blood vessels [40]. Onorato et al. provided insights into the mechanism of action of pyridoxamine as an AGE inhibitor and suggested that pyridoxamine may help inhibit the increased chemical modification of tissue proteins in diabetes, atherosclerosis, and other chronic diseases, thereby alleviating age-related diseases [41]. Additionally, pyridoxamine reduces superoxide radicals produced by H_2_O_2_, possibly through a scavenging mechanism and reducing lipid peroxidation in cells [42]. Our data indicate that pyridoxamine shows a declining trend with age in pigs, thereby increasing the likelihood of diseases such as diabetes during the aging process. The reduction in its content increases the production of free radicals, accelerating the aging process.

In addition, according to the KEGG analysis based on differential metabolites, pyrimidine metabolism is a significant differential profile common to all three groups. Cytidine-5-monophosphate, a product in its metabolic process, is a nucleotide within the cell that participates in many important biochemical processes, including nucleic acid synthesis, phospholipid metabolism, and phosphorylation reactions [43], playing an important role in the immunity of some adults, infants, and young mammals [44]. Notably, Nakagawara et al. reported that cytidine-5-monophosphate enhances the expression of PGC-1α in mouse C2C12 cells and promotes the formation of myotubes [45], and PGC-1α can protect skeletal muscles and alleviate the occurrence of muscle atrophy [46], which may suggest a molecular mechanism for muscle atrophy during the aging process in humans or animals. During the aging process, a series of changes occur in cellular metabolism and function, which may affect the level and role of cytidine-5-monophosphate. However, there is currently no direct evidence to suggest that cytidine-5-monophosphate itself promotes or delays the aging process. Our study indicates that cytidine-5-monophosphate consistently shows a declining trend with age, suggesting its potentiality as a biomarker involved in the biological aging process, although its specific mechanism remains to be investigated. However, it is important to acknowledge the limitations of this study. The correlation between the identified metabolites and blood metabolite profiles remains unexplored, which could provide a more comprehensive understanding of metabolic changes during aging. Furthermore, the validation of the data through independent cohorts or longitudinal studies is essential to confirming the robustness of these findings.

## 5. Conclusions

In conclusion, this study has revealed significant metabolic variations across age groups in pigs, as demonstrated through comprehensive metabolomic analysis. Notable metabolites, such as cholecalciferol, MetO, and pyridoxamine, exhibited consistent patterns of fluctuation with age, providing critical insights into the aging process in these animals. Importantly, the findings highlight the pyrimidine metabolic pathway, particularly its product cytidine-5-monophosphate, as a promising biomarker for aging-related research. These insights have profound implications for both livestock management and our understanding of animal and human aging. By elucidating the regulatory mechanisms driving these metabolic changes, future research can elucidate their roles in aging and related diseases, potentially leading to innovative strategies for intervention and treatment. The integration of metabolic profiling in aging studies could pave the way for breakthroughs in both veterinary and medical fields.

## Figures and Tables

**Figure 1 metabolites-14-00558-f001:**
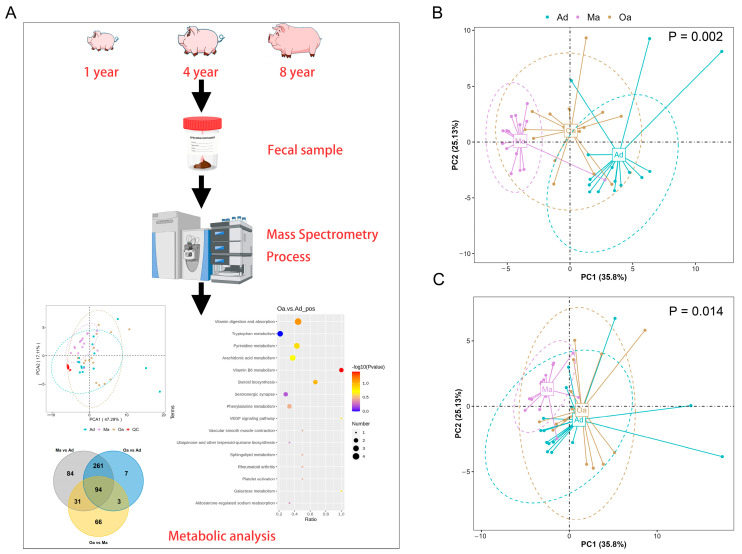
The overall changes in metabolomics among the pigs in the Ad, Ma, and Oa groups. (**A**) A schematic representation of the sample preparation process and the steps involved in metabolomic profiling and analysis is provided. (**B**,**C**) Principal component analysis (PCA) was conducted based on the relative abundances of metabolites across the three groups, presented in positive ion mode (**B**) and negative ion mode (**C**), a *p*-value of less than 0.05 indicates that there are significant differences in the composition of the fecal metabolome among pigs of different age groups.

**Figure 2 metabolites-14-00558-f002:**
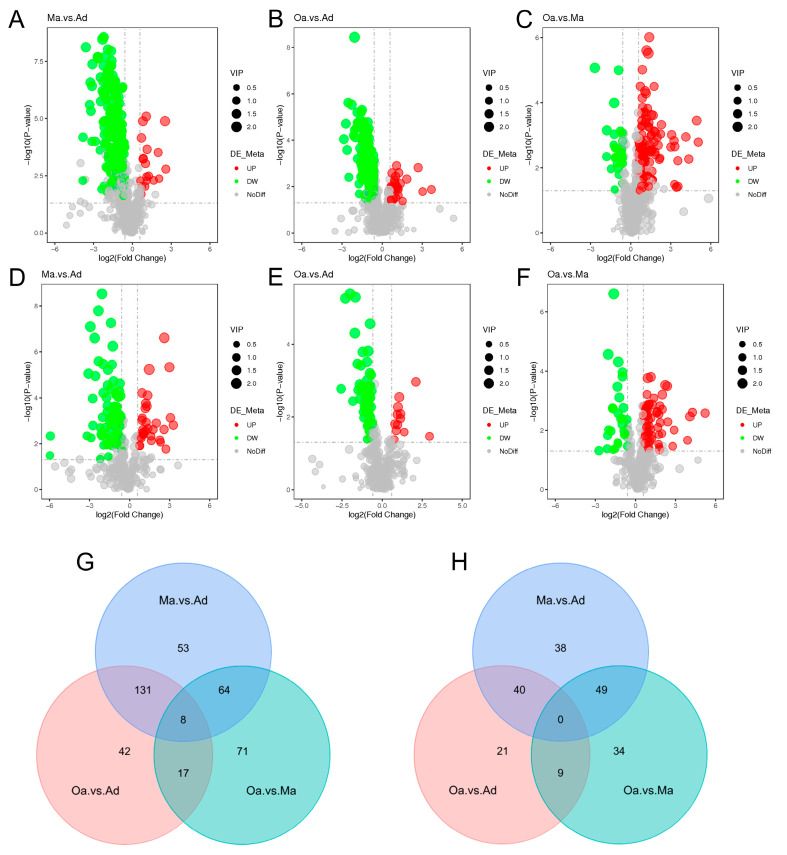
The overall and dynamic changes in stool metabolomics associated with aging. (**A**–**F**) Volcano plots highlight the fecal metabolites that were increased (red) or decreased (green) in the comparison group of Ma versus Ad, Oa versus Ad, and Oa versus Ma in positive ion mode (**A**–**C**) and negative mode (**D**–**F**), with FDR < 0.05, log2 fold change (FC) >0.25 or <−0.25. (**G**) The Venn diagram illustrated the intersection based on significant differential fecal metabolites derived from the comparison of any two groups, and revealing that 8 metabolites were present in the differential results of all 3 pairwise comparisons in positive ion mode. (**H**) No intersection was found in negative ion mode.

**Figure 3 metabolites-14-00558-f003:**
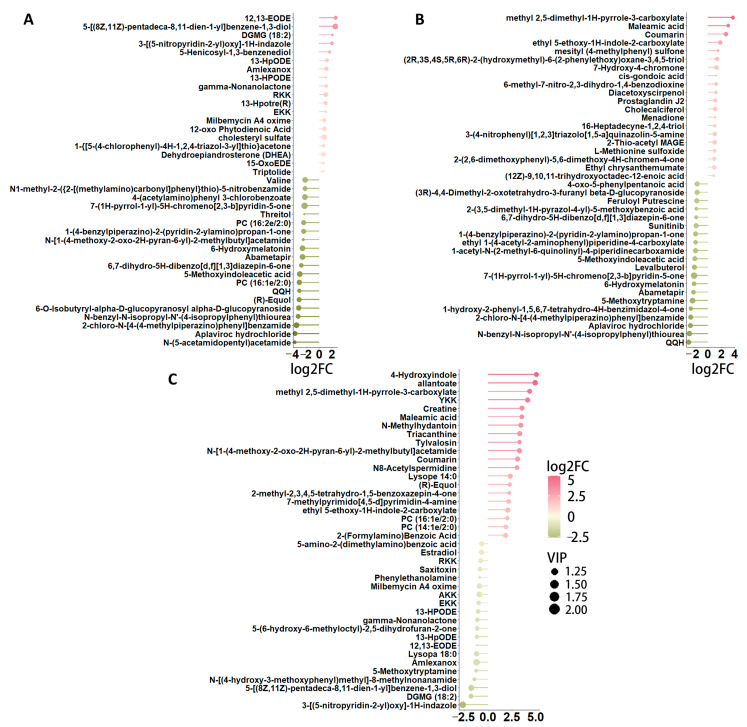
A detailed examination of the differences in metabolite profiles across various age groups. (**A**) Stem plot shows the top 39 significant different metabolites between the pig in middle-aged (Ma) and one-year-old (Ad) groups, highlighting significant metabolic variations that may reflect age-related physiological changes. (**B**) A similar comparison between the elderly (Oa) and one-year-old (Ad) groups is shown by the stem plot, further elucidating the metabolic alterations that occur as pigs transition into later stages of life. Finally, panel (**C**) illustrates the differences in metabolites between the elderly (Oa) and middle-aged (Ma) groups, offering insights into the metabolic pathways that may be uniquely affected during this critical period of aging.

**Figure 4 metabolites-14-00558-f004:**
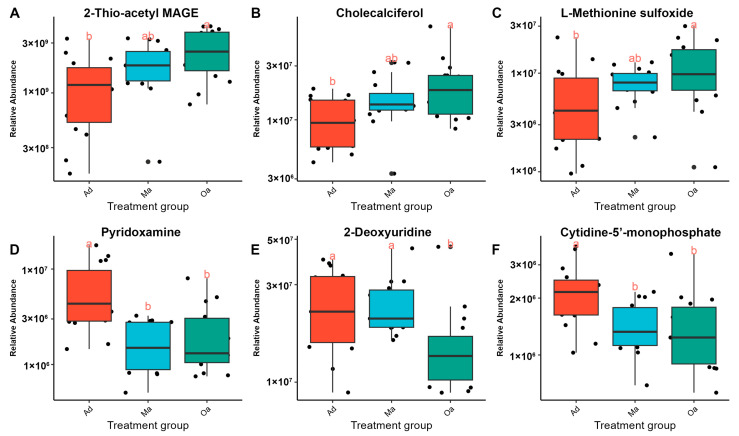
A panel of metabolites changes associated with aging. Panels (**A**–**C**) display box plots illustrating the up-regulated metabolites observed with advancing age. These metabolites demonstrate significant increases, highlighting their potential role in the physiological adaptations that occur as pigs age. Conversely, panels (**D**–**F**) depict box plots for down-regulated metabolites, which exhibit notable decreases in concentration with age. (Distinct lowercase letters above the boxplot represent significant differences in metabolite changes among different age groups (*p* < 0.05)).

**Figure 5 metabolites-14-00558-f005:**
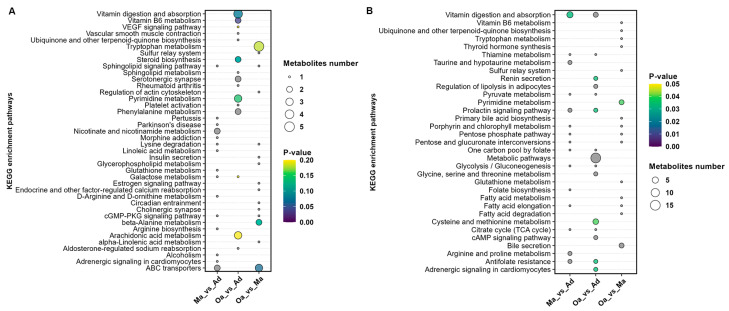
Illustration of the results of a metabolic pathway analysis conducted based on metabolites that exhibited differences between pairwise groups in both positive ion modes. (**A**) and negative ion mode (**B**). The size of the circles is indicative of the degree of pathway enrichment, with larger circles representing pathways that are more significantly enriched. Additionally, the color of the circles reflects the statistical significance of each pathway, providing a visual representation of the relevance of these metabolic alterations across the age groups analyzed. This dual-mode analysis enhances the understanding of the intricate metabolic changes associated with aging in pigs, facilitating the identification of key pathways that may warrant further investigation in future studies.

## Data Availability

Data are available on reasonable request.

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
