# Peer review of "Unveiling the Metabolic Trajectory of Pig Feces Across Different Ages and Senescence"

_metabolites, 2024, doi:10.3390/metabo14100558_

Round 1

Reviewer 1 Report

Comments and Suggestions for Authors

This is an interesting manuscript on the biology of aging, wherein pig has been employed to study the metabolite related changes keeping in view its relevance to the biological model fore humans. However, the manuscript requires major changes with respect to introduction, methodology and discussion. Also the major lacuna in this study is lack of correlation of blood metabolite data, which could have provided a clear picture of age related changes in metabolomic profile in pig.

Abstract:

1.       The abstract and the introduction to the manuscript does not go in sync. The abstract should be representative of the manuscript, however, the background/objective as depicted in abstract (“pig centered”) and the background/objectives/aim of study as described in the introduction (“human centered”) is different. Authors should change either change the abstract background or background of the Introduction.

2.       The conclusion of abstract is very general, authors should include the name of the senescence biomarkers they obtained in this study.

Introduction:

3.       Lines (26-44) - The authors should decrease the overemphasis on the background of study depicting aging in humans.  It would be better if the focus is on pig production and health.

Methodology

4.       Line (78-82)- The pig population details should be included like body weight, breed, sex etc.

5.       Line (92-96)- The process/method of sample preparation and UHPLC-MS/MS methods should be detailed, either in text or as supplementary material.

Discussion

6.       Line 246 – Delete repeated words.

7.       Line 248-250 – This line should be removed. “The aim is to identify ….”. It seems redundant here.

8.       Line 252- “either upregulated or downregulated, with increasing age”. It is confusing. The metabolites should be specified for up or down regulation.

9.       Line (255-280) - The discussion is so general and human centric. Authors may refer to studies in humans, but they should discuss the findings with respect to metabolism and physiology of pigs.

10.    Line (281-300) – While discussing the metabolite role/function, the authors should indicate, if they are up or downregulated and their implication for aging process.

11.   The weakness of the study like correlation with blood metabolite profile and validation of the data is missing, should be discussed.

Conclusion

12.   Line (399-342) – This conclusion is very general. It should be either removed or revised.

Author Response

Editors,

Metabolites

September 23, 2024

Dear Editors and Reviewers:

We feel grateful for your letter and we appreciate the reviewers for their thoughtful and constructive comments concerning our manuscript entitled “Unveiling the Metabolic Trajectory of Pig Feces across Different Ages and Senescence” (Manuscript ID: metabolites-3211937), which have guided the revision of this manuscript. Major revisions in the manuscript have been highlighted in blue for easy identification. The concerns have been addressed as follows:

# Reviewer 1

This is an interesting manuscript on the biology of aging, wherein pig has been employed to study the metabolite related changes keeping in view its relevance to the biological model for humans. However, the manuscript requires major changes with respect to introduction, methodology and discussion. Also the major lacuna in this study is lack of correlation of blood metabolite data, which could have provided a clear picture of age related changes in metabolomic profile in pig.

Abstract:

  1. The abstract and the introduction to the manuscript does not go in sync. The abstract should be representative of the manuscript, however, the background/objective as depicted in abstract (“pig centered”) and the background/objectives/aim of study as described in the introduction (“human centered”) is different. Authors should change either change the abstract background or background of the Introduction.

Response: We sincerely appreciate your thorough examination and valuable suggestions regarding this work. To ensure alignment with the descriptions provided in the introduction, the abstract in our manuscript has been carefully reviewed and revised.

  1. The conclusion of abstract is very general, authors should include the name of the senescence biomarkers they obtained in this study.

Response: We agree and have corrected this.

Introduction:

  1. Lines (26-44) - The authors should decrease the overemphasis on the background of study depicting aging in humans.  It would be better if the focus is on pig production and health.

 Response: We really appreciate the reviewer’s comments. We have revised this part of contents.

Methodology

  1. Line (78-82)- The pig population details should be included like body weight, breed, sex etc.

Response: Thanks for this suggestion, and we have added the abovementioned information.

  1. Line (92-96)- The process/method of sample preparation and UHPLC-MS/MS methods should be detailed, either in text or as supplementary material.

Response: Revised as suggested.

Discussion

  1. Line 246 – Delete repeated words.

Response: Revised as suggested.

  1. Line 248-250 – This line should be removed. “The aim is to identify ….”. It seems redundant here.

Response: Revised as suggested.

  1. Line 252- “either upregulated or downregulated, with increasing age”. It is confusing. The metabolites should be specified for up or down regulation.

Response: Revised as suggested.

  1. Line (255-280) - The discussion is so general and human centric. Authors may refer to studies in humans, but they should discuss the findings with respect to metabolism and physiology of pigs.

Response: Thanks for the reviewer’s comment. We have rephrased these sentences to make them more rational in accordance to the topic of this manuscript.

  1. Line (281-300) – While discussing the metabolite role/function, the authors should indicate, if they are up or downregulated and their implication for aging process.

Response: Thank you for your insightful comments regarding the discussion of metabolite roles and their regulation in the aging process. The importance of clearly indicating whether metabolites are upregulated or downregulated, along with their implications for aging, is greatly appreciated. In response to your feedback, the manuscript has been revised to include a detailed discussion of the roles and functions of relevant metabolites in the context of porcine aging.

  1. The weakness of the study like correlation with blood metabolite profile and validation of the data is missing, should be discussed.

Response: Thanks for this valuable comment and we have revised accordingly.

Conclusion

  1. Line (399-342) – This conclusion is very general. It should be either removed or revised.

Response: Thanks for this comment, we have revised the contents in conclusion.

Reviewer 2 Report

Comments and Suggestions for Authors

Dear authors,

I will recommend your manuscript for publication, but I has some questions for edition.

L78 – Please add sex of investigated animals and season of sample collecting.

L84 – Term “UHPLC” need to be descripted first time.

L101 – Add links to all databases.

L128 – I did not see PCA in Statistical Methods

L128 – Hou are you count significant separation in PCA?

L187 – Figure not informative – too small font.

L211 – KEGG Enrichment need to be manifested in Methodes.

Regards,

Author Response

Editors,

Metabolites

September 23, 2024

Dear Editors and Reviewers:

We feel grateful for your letter and we appreciate the reviewers for their thoughtful and constructive comments concerning our manuscript entitled “Unveiling the Metabolic Trajectory of Pig Feces across Different Ages and Senescence” (Manuscript ID: metabolites-3211937), which have guided the revision of this manuscript. Major revisions in the manuscript have been highlighted in blue for easy identification. The concerns have been addressed as follows:

# Reviewer 2

Dear authors,

I will recommend your manuscript for publication, but I has some questions for edition.

Response: We really appreciate the reviewer’s comments. We re-corrected the manuscript as advised by you.

L78 – Please add sex of investigated animals and season of sample collecting.

Response: Revised as suggested.

L84 – Term “UHPLC” need to be descripted first time.

Response: Revised as suggested.

L101 – Add links to all databases.

Response: Thank you for your valuable comments. To enhance the accessibility and utility of the article, direct links to all referenced databases have been incorporated into the revised version. These links now guide readers to the corresponding database pages, allowing them to readily access additional information and data.

L128 – I did not see PCA in Statistical Methods

Response: We have added the description of PCA in the part of Statistical Methods.

L128 – Hou are you count significant separation in PCA?

Response: Thanks for this comment. The significant separation among the different age groups was evaluated using PERMANOVA analysis, and we have rephrased the relevant sentences in the revised manuscript.

L187 – Figure not informative – too small font.

Response: We appreciate the editor's comments and have redrawn Figures, enhancing the image resolution and readability.

L211 – KEGG Enrichment need to be manifested in Methods.

Response: Thanks for this comment, and we have revised the methods that related to KEGG Enrichment analysis, we describe each step of KEGG enrichment analysis in detail, including the statistical methods used, the threshold settings, and the interpretation of the results. We believe that these additions will help readers to better understand the results of KEGG enrichment analysis.

Round 2

Reviewer 1 Report

Comments and Suggestions for Authors

Most of the suggestions have been incorporated and the manuscript has been substantially revised.

Line 273-284 and elsewhere in manuscript: However, authors need not to capitalize the "metabolite names" in the middle of the paragraph. E.g. "L-Methionine sulphoxide" can be written as "L-methionine sulphoxide"; , "Cysteine" can be written as "cysteine". Uniformity should be maintained throughout the manuscript.  Similarly for " Cholecalciferol" use "cholecalciferol".

Line 328 and elsewhere: Use abbreviation, (MetO). The authors should edit the manuscript on similar lines for other abbreviations.  

Line 334 and 341: What does AD refers to and how does it relate to pigs. aquthors should first introduce the term before abbreviating it and then use the abbreviations in the manuscript. Also, the authors should be rational in discussing the points, while refering to human diseases/pathologies in pigs. Revision in lines 332-346 are not stisfactory. Thw authors should re-revise the contents.

The conclusion has been revised, however, it still remains general. The authors can remove " Lines 392-395". Conclusion should present the main findings of the study; overemphasis on general points should be avoided. 

Figure 4- Legend and elsewhere : The authors should include level of significance, where ever applicable.

Author Response

Editors,

Metabolites

September 27, 2024

Dear Editors and Reviewers:

We feel grateful for your letter and we appreciate the reviewers again for their thoughtful and constructive comments concerning our manuscript entitled “Unveiling the Metabolic Trajectory of Pig Feces across Different Ages and Senescence” (Manuscript ID: metabolites-3211937), which have guided the revision of this manuscript. Minor revisions in the manuscript have been highlighted in red for easy identification. The concerns have been addressed as follows:

# Reviewer’s opinions

Most of the suggestions have been incorporated and the manuscript has been substantially revised.

Line 273-284 and elsewhere in manuscript: However, authors need not to capitalize the "metabolite names" in the middle of the paragraph. E.g. "L-Methionine sulphoxide" can be written as "L-methionine sulphoxide"; , "Cysteine" can be written as "cysteine". Uniformity should be maintained throughout the manuscript.  Similarly for " Cholecalciferol" use "cholecalciferol".

Response: Thanks for the reviewer’s valuable suggestion. We have revised the metabolite names with lowercase the first letter throughout the manuscript.

Line 328 and elsewhere: Use abbreviation, (MetO). The authors should edit the manuscript on similar lines for other abbreviations.  

Response: Thanks for this comment, and we have revised it accordingly.

Line 334 and 341: What does AD refers to and how does it relate to pigs. authors should first introduce the term before abbreviating it and then use the abbreviations in the manuscript. Also, the authors should be rational in discussing the points, while refering to human diseases/pathologies in pigs. Revision in lines 332-346 are not stisfactory. The authors should re-revise the contents.

Response: Thanks for this comment. We have revised it accordingly, please see the new manuscript.

The conclusion has been revised, however, it still remains general. The authors can remove " Lines 392-395". Conclusion should present the main findings of the study; overemphasis on general points should be avoided. 

Response: Thanks for this advice, we have deleted the inappropriate sentences.

Figure 4- Legend and elsewhere: The authors should include level of significance, where ever applicable.

Response: Thanks for pointing out this defect, we revised them as suggested.